# Proprietary Algorithms for Polygenic Risk: Protecting Scientific Innovation or Hiding the Lack of It?

**DOI:** 10.3390/genes10060448

**Published:** 2019-06-13

**Authors:** A. Cecile J.W. Janssens

**Affiliations:** Department of Epidemiology, Rollins School of Public Health, Emory University, 1518 Clifton Road NE, Atlanta, GA 30322, USA; cecile.janssens@emory.edu; Tel.: +1-404-727-6307

**Keywords:** personal genomics, DNA, polygenic, risk, regulation, discrimination, calibration, prediction, transparency, autonomy

## Abstract

Direct-to-consumer genetic testing companies aim to predict the risks of complex diseases using proprietary algorithms. Companies keep algorithms as trade secrets for competitive advantage, but a market that thrives on the premise that customers can make their own decisions about genetic testing should respect customer autonomy and informed decision making and maximize opportunities for transparency. The algorithm itself is only one piece of the information that is deemed essential for understanding how prediction algorithms are developed and evaluated. Companies should be encouraged to disclose everything else, including the expected risk distribution of the algorithm when applied in the population, using a benchmark DNA dataset. A standardized presentation of information and risk distributions allows customers to compare test offers and scientists to verify whether the undisclosed algorithms could be valid. A new model of oversight in which stakeholders collaboratively keep a check on the commercial market is needed.

## 1. Introduction

In the early days of genome-wide association studies (GWAS) [1], companies like 23andMe, Navigenics, and deCODEme started offering risk predictions for multiple complex diseases from a single DNA scan. With no more than a dozen single nucleotide polymorphisms (SNPs) identified for each disease, these offers seemed overoptimistic and premature. When we started to investigate the predictive ability of these algorithms, we discovered that none of these companies disclosed the details of their risk calculations online [2]. To study predictive ability, we had to reconstruct the algorithms using the information about the statistical models, included SNPs, and disease risks from the citations provided on the companies’ websites [2]. 

Our study showed that, at the population level, the algorithms of 23andMe, Navigenics, and deCODEme predicted with similar accuracy. The predictive ability differed between diseases: age-related macular degeneration and Crohn’s disease, for example, were predicted with high accuracy (area under the receiver-operating characteristic curve (AUC) 0.75–0.80), but the prediction of atrial fibrillation was barely better than tossing a coin (AUC 0.58–0.62) [2]. Despite their comparable predictive ability at the population level, we found that the differences in the risk predictions for individual customers were unacceptably large. 

Figuring out the algorithms for six diseases from three companies was a time-consuming project. Two of the three companies had already left the market before we were ready to submit our study for publication. 

When predicted risks depend on who makes the prediction [2,3,4], customers should demand more insight into how the polygenic risk algorithms are constructed and evaluated. For a market that thrives on the premise that customers should be able to make their own decisions about genetic testing, it is disconcerting that customers are barely informed about the rigor and rationale behind the calculations. 

Companies keep algorithms proprietary to maintain a competitive advantage. This allows them to be first, to acquire a market share, and to improve their algorithms before others do. Keeping algorithms proprietary, as trade secrets, is protected under the law [5]. We should anticipate that companies have little incentive to voluntarily disclose algorithms for as long as the algorithms, rather than the service or price, are essential for competitive advantage. Yet, even when algorithms are kept proprietary, there is a lot that scientists do know about them and a lot that companies can disclose to increase the credibility of their algorithms without sharing them. 

For example, prediction algorithms are typically built from known predictors or risk factors that epidemiological studies had identified earlier and that are combined using known statistical models. It is unlikely that companies use SNPs, rare mutations, gene–gene interactions, or other mechanisms that are unknown to scientists and that substantially improve the predictive ability of their tests without first taking scientific credit for the discovery. Thus, based on the genetic epidemiological literature, researchers have expectations about what the algorithms might look like and how predictive they might be [2]. 

Furthermore, in 2011, a group of experts published guidelines to strengthen the reporting of genetic risk prediction studies [6], identifying 25 reporting items that address all aspects that are deemed essential for understanding how prediction algorithms are developed and evaluated. Disclosing the algorithm is only one piece of the information that companies can share with customers and the public to provide insight into the quality of the predictions. Companies should be encouraged to disclose everything else.

## 2. What Insight Should Customers Demand? 

When customers want to make an informed decision about whether to purchase a personal genome test, they should want to know whether the test can give them the answer they are seeking. Can the algorithms in the test predict risks that they are interested in? It is not evident that all tests can do this. 

Knowing the predictive ability is important because most polygenic algorithms cannot predict risks from 0% to 100%. For example, a polygenic risk score for coronary artery disease (CAD) using up to 6.6 million SNPs predicted an 11% risk for the people in the top 1% of the distribution and 0.8% risk for those in the lowest 1% [7]. The probability of not developing CAD for the people in the 1% upper tail thus is still 89%. In the same study, the average predicted risks for the top 1% of the distribution was approximately 13% for breast cancer, 6% for atrial fibrillation and type 2 diabetes, and 4% for inflammatory bowel disease. The height of the risks in the upper tail of the distribution is expected to be higher when the predictive ability is higher and the disease is more common. 

Figure 1 illustrates four risk distributions with different predictive abilities for a hypothetical disease that affects 20% of the population. A test that is unable to predict the disease will predict the same 20% risk for everyone (Figure 1a): this test is unable to discriminate between people who will develop the disease and those who will not. At the other extreme is a perfect test that predicts a 100% risk for 20% of the people who will develop the disease and a 0% risk for all others (Figure 1b). This test has a perfect discriminative ability. The predictive ability of polygenic algorithms will present as distributions across a variable range of predicted risks (Figure 1c,d). The higher the predictive ability, the more spread there is in the distribution of risks and the higher the AUC. The AUC ranges from 0.50 (Figure 1a) to 1.0 (Figure 1b).

The risk distributions are only informative if we can assume that the algorithms are well-calibrated. Calibration indicates whether predicted risks match observed risks [8], whether for example, of all people with a 20% risk of disease, 20% will develop the disease and not, say 14% or 25%. It cannot be assumed that commercial algorithms are well-calibrated. Companies typically construct polygenic algorithms using literature data for the weights of the SNPs and the population disease risks. As this data may vary between studies, it is essential to verify whether the data used for the algorithms was appropriate. 

Calibration is the Achilles’ heel of commercial polygenic algorithms. The assessment of calibration requires a prospective cohort study in which a population at risk of the disease is genotyped and followed over time to monitor who develops the disease. Without prospective cohort data, researchers can only publish distributions of polygenic scores but not distributions of polygenic risks. The two distributions are related—the higher the polygenic score, the higher the risk—but polygenic scores cannot be interpreted as risks. Without verification in prospective cohort data, the calibration of predictive algorithms remains unknown and a concern. 

Discrimination and calibration are two essential indicators of the predictive ability of an algorithm. When a well-calibrated algorithm predicts that a customer’s risk is 20%, then it is 20% irrespective of the predictive ability of the test. If the risk was predicted using a test that cannot discriminate (Figure 1a), then their risk was and is 20% like everyone else’s—the genetic test was unable to change the predicted risk. 

Discrimination and calibration are essential but cannot tell whether predicted risks are relevant for an individual, i.e., whether a risk is their risk. The relevance depends, among other factors, on the population for which an algorithm was developed. Many risks of disease are higher among older people or men, and the predictive ability of polygenic algorithms is known to vary between ethnicities [9,10]. Verifying if a customer could have been a member of the study population for which the algorithm was constructed tells whether the risk could apply to them [11].

As predicted risks are more informative when they are estimated using an algorithm that can predict a wider range of risks, the value of learning about one’s risk depends on the risk estimates that others might get. Therefore, customers should be interested in knowing the expected distribution of risk when considering the purchase of a test—knowing the risk distribution before buying informs whether the test can predict risks that a customer would find meaningful enough to be worth buying the test. In our example, if a customer is interested to learn whether their risk of disease is 50% or higher, they can see from the risk distribution that the algorithm in Figure 1c is unable to predict such high risks. The algorithm in Figure 1d can predict risks over 50%, but these are exceptions. Estimates of the percentage of people that is expected to have a risk higher than a certain risk (here, 50%) are presented in cumulative risk distributions. 

## 3. Toward Transparency and Accountability

When the United States Food and Drug Administration (FDA) granted 23andMe permission to market its Personal Genome Service Test, the agency required that the company adopt ‘special controls’ for the information that customers need to receive before purchase and before and after testing [12]. The FDA specified in its letter that this information should include “the health condition/disease being tested, the purpose of the test, the information the test will and will not provide, the relevance of race and ethnicity to the test results, information about the population to which the variants in the test are most applicable, the meaning of the result(s), other risk factors that contribute to disease, appropriate follow-up procedures, how the results of the test may affect the user’s family, including children, and links to resources that provide additional information” [12]. This information should help the customer to find out whether the test is relevant and useful to them: e.g., do they belong to the population for whom the test was developed, understand what the test results might mean, and find the follow-up procedures useful? 

The FDA also asked insight in the predictive ability of the ‘algorithms.’ As the risk assessments for which 23andMe asked permission were simple, up to a few variants for each of the ten diseases, the FDA requested that the company report the likelihood ratios for all variants or variant combinations [12], which is the risk of disease for people with the variant combination relative to the general population. The FDA letter explained how 23andMe could calculate the likelihood ratio from the odds ratio and the pre-test risk of disease.

When the FDA asks for likelihood ratios of all variants or variant combinations, the agency essentially asks for disclosure of the algorithms. The likelihood ratios can be transformed into predicted risks using Bayes’ theorem [13], which together with the frequency of the variant combinations allows constructing the risk distribution. The letter of 23andMe to the FDA did provide this information [14], but the company’s website and the “23andMe Personal Genome Service Genetic Health Risk Reports V5 Package Insert” do not; they report most of the required ‘special controls’ but give no information about the odds ratios or likelihood ratios of the variant combinations [15].

Asking the likelihood ratios for variant combinations is undoable when polygenic risk scores are constructed from many SNPs. If the FDA wants full disclosure for polygenic algorithms, the agency should ask the code, including the pre-test risks of disease by combinations of age, sex, ethnicity, and other relevant predictors, if pertinent. If the agency wishes to honor requests to keep algorithms proprietary, it should ask companies to share information about the logic behind their algorithms [16], details about the construction and validation of the algorithm and information about the expected distribution of predicted risks. In a recent white paper, 23andMe discloses how the company developed and evaluated its new algorithm for the prediction of type 2 diabetes [17]. While there is room for improvement and for translating the essence of the white paper in plain language to customers, this paper sets an example for its rigor and transparency.

This distribution can be obtained by applying the algorithm to DNA data from a large repository or existing cohort that is deemed suitable for this purpose. There is ample evidence that assessing the predictive ability in simulated or hypothetical data yields valid approximations of real-life data [13]. When companies are required to provide information about the development of their algorithms and present risk distributions from the application of their algorithms to the same data, customers can compare the test offers between companies. Scientists can use this information to verify whether the presented distributions are what could be expected on the state of scientific knowledge and aim to develop and compare transparent, evidence-based algorithms [18]. 

Given the fast progress in genomics research and frequent updates in prediction algorithms, it is unrealistic to expect that the FDA can enforce the regulations in a timely fashion. A new governance model is needed where the FDA “could require as a condition of marketing that developers disclose how the algorithm was developed, the data used for that development, and—to the extent known—how the algorithm works” and “could also mediate information sharing of post-market-surveillance information” [19]. Gathering evidence for the clinical validity of algorithms should be a collaborative responsibility for providers, healthcare institutes, insurers, and customer organizations.

## 4. Concluding Remarks

The *raison d’être* of direct-to-consumer genetic testing is that customers, not doctors, decide whether they are interested in taking a test and finding utility in testing. If companies truly respect their customers and honor their ability to make autonomous and informed decisions about genetic testing, then they should provide as much detail about the tests as possible while allowing algorithms to remain proprietary.

Given that risk algorithms are not completely black box algorithms—there is much scientific evidence about the predictive ability and potential of polygenic risk scores—, it is in the companies’ interests to disclose their rigorous approaches to the development of valid and relevant algorithms. They should share the logic and key sources and decisions in the development of the algorithms and offer insight in the expected risk distributions stratified by age, sex, and ethnicity, tailored to the specific characteristics of the customer. A standardized format for sharing this information helps encourage transparency and comprehension. 

This transparency about the direct-to-consumer polygenic algorithms should be requested for customers but also for providers and insurers. Customers pay for the DNA tests, but that price does not include the costs that follow when they visit their physician for examination of their ‘high’ genetic risks. If the predictive ability of commercial algorithms is poor, many people with high predicted risks will visit their doctors unnecessarily [20,21]. This is why, to protect the affordability of healthcare, companies should only recommend visiting a health professional when their referral based on the test result is in line with medical protocols or guidelines [22]. 

Oversight of the direct-to-consumer market for polygenic risk algorithms is complex and time-sensitive. Algorithms are frequently adapted to the latest scientific insights, which may make evaluations obsolete before they are completed. A standardized format for the provision of essential information could readily provide insight into the logic behind the algorithms, the rigor of their development, and their predictive ability. The development of this format gives responsible providers the opportunity to lead by example and show that much can be shared when there is nothing to hide.

## Figures and Tables

**Figure 1 genes-10-00448-f001:**
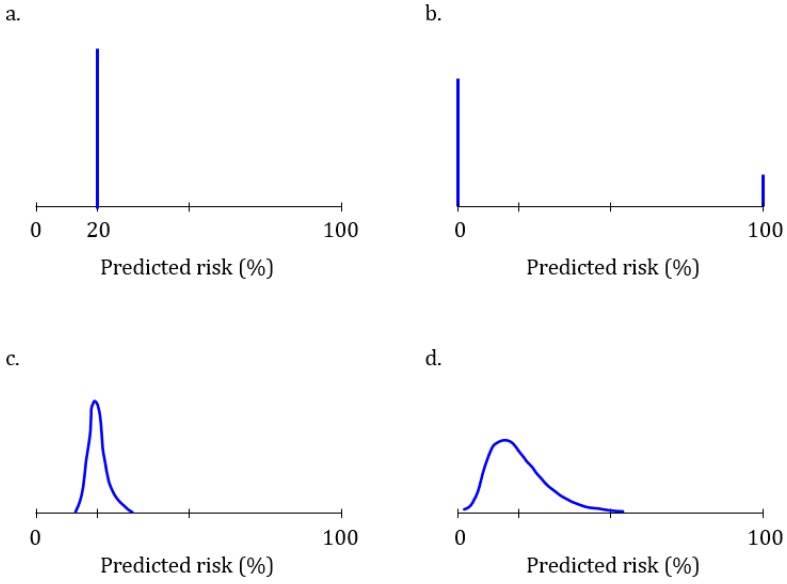
Risk distributions of four different polygenic algorithms for a hypothetical disease that affects 20% of the population. (**a**) Uninformative polygenic algorithm that predicts a 20% risk of disease for everyone; (**b**) Perfect polygenic algorithm that predicts a 100% risk for the 20% of the people who will develop the disease and 0% risk for the other 80%; (**c**,**d**) Polygenic algorithms with lower (**c**) and higher (**d**) predictive ability.

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
