# Peer review of "Proprietary Algorithms for Polygenic Risk: Protecting Scientific Innovation or Hiding the Lack of It?"

_genes, 2019, doi:10.3390/genes10060448_

Round 1
Reviewer 1 Report
Your manuscript calls for for more transparency from genetic testing companies on how their risk algorithms are calculated and what they mean for individuals taking the test. I commend the sentiments expressed throughout and I particularly like the suggestion that consumers should be empowered with knowledge of different tests to allow them to make decisions. However, as obtaining details of proprietary algorithms has proven difficult for commercial reasons, it may be more prudent to ask to companies to release the details of their algorithms to an independent (regulatory) body who will calculate variables such as risk distributions stratified for age, sex etc without releasing the algorithm itself.
Minor typos:
Line 34: Crohn's not Crohn
Line 34: could you provide a ballpark figure for 'hifh accuracy'?
Author Response
I agree that sharing the algorithm with an independent regulatory body would be ideal, but companies have no obligation to share proprietary 'trade secrets'. I have asked and received feedback on this paper from 23andMe, and found that the company had shared the requested information with the FDA, just not with the customers. It is likely that the information shared with the FDA is already outdated as companies keep improving their algorithms.
- typo corrected
- information about high accuracy is now provided. Because this is information about the area under the ROC curve, I have now added an explanation about that too, including linking AUC to the graphs.
Reviewer 2 Report
This is a cogent commentary on a timely topic that will be of great interest to the genetics and tech communities. It is well written in addition to being well reasoned, and I have no substantive edits to suggest.
Author Response
Thank you!